# Update on Biogenic Amines in Fermented and Non-Fermented Beverages

**DOI:** 10.3390/foods11030353

**Published:** 2022-01-26

**Authors:** Pierina Visciano, Maria Schirone

**Affiliations:** Faculty of Bioscience and Technology for Food, Agriculture and Environment, University of Teramo, Via R. Balzarini, 1, 64100 Teramo, Italy; pvisciano@unite.it

**Keywords:** biogenic amines, alcoholic beverages, wine, beer, fruit juices, plant drinks

## Abstract

The formation of biogenic amines in food and beverages is mainly due to the presence of proteins and/or free amino acids that represent the substrates for microbial or natural enzymes with decarboxylation or amination activity. Fermentation occurring in many alcoholic beverages, such as wine, beer, cider, liqueurs, as well as coffee and tea, is one of the main processes affecting their production. Some biogenic amines can also be naturally present in some fruit juices or fruit-based drinks. The dietary intake of such compounds should consider all their potential sources by both foods and drinks, taking in account the health impact on some consumers that represent categories at risk for a deficient metabolic activity or assuming inhibiting drugs. The most important tool to avoid their adverse effects is based on prevention through the selection of lactic acid bacteria with low decarboxylating activity or good manufacturing practices hurdling the favoring conditions on biogenic amines’ production.

## 1. Introduction

The biogenic amines (BAs) are nitrogenous compounds deriving from enzymatic reactions such as decarboxylation, transamination, reductive amination, and degradation of the corresponding precursor amino acids. Based on their chemical structure, they are distinguished as aliphatic (cadaverine, putrescine, agmatine, ornithine, spermidine, and spermine), aromatic (β-phenylethylamine and tyramine), or heterocyclic (tryptamine and histamine). They are also classified into monoamine (tyramine and β-phenylethylamine), diamine (histamine, serotonin, tryptamine, putrescine, and cadaverine), and polyamine (agmatine, spermine, and spermidine) according to their number of amine groups [1]. In Table 1, the formula and chemical structure of most BAs are shown.

The availability of free amino acids, the presence of decarboxylase-positive microorganisms, and further conditions favoring the microbial load development, such as an improper temperature of storage and ripening or fermentation processes, can increase their production. Microorganisms belonging to specific genera, such as *Pseudomonas*, *Clostridium*, *Bacillus*, and *Photobacterium*, as well as to the *Enterobacteriaceae* family (i.e., *Escherichia*, *Klebsiella*, *Citrobacter*, *Proteus*, *Shigella*, and *Salmonella* genera), are reported as BA sources. In fermented products, such as cheese, sausages, and fermented vegetables, but also beverages, lactic acid bacteria (LAB) produce BAs to survive on an acidic milieu during fermentation. The genera *Enterococcus*, *Lactococcus*, *Lactobacillus*, *Pediococcus*, *Carnobacterium*, and *Leuconostoc* can decarboxylate amino acids in foods and drinks [2].

Some conditions occurring during the fermentation process of foods and drinks can significantly affect and/or increase the synthesis of BAs in the final product. It has been demonstrated that the decarboxylation of amino acids is more active in acid environments where pH ranges from 4.0 to 5.5 and when the temperature is between 20 °C and 37 °C [3].

An opposite situation is observed with temperature, i.e., while lower pH correlates with higher BA content, lower temperatures reduce both bacterial growth and decarboxylase activity, and BA production significantly decreases below 5 °C [3]. Additionally, in beverages, the BA amounts are influenced by the availability of free amino acids from the proteolytic activity, as well as by the pH and temperature of storage. The use of sulfur dioxide (SO_2_) in wines can also reduce both microbial and enzymatic activities [4].

The main source of exogenous BAs is the diet through the uptake of foods or beverages containing high concentrations of these compounds. The BAs profile in beverages varies according to the raw material, endogenous enzymes, and microorganisms naturally present or added during the production process. In this review, the main BAs occurring in fermented and non-fermented beverages, their concentrations reported in literature, and the prevention measures able to reduce their presence will be described.

## 2. Biogenic Amines in Fermented Alcoholic Beverages

The alcoholic beverages are characterized by a fermentation process with production of ethanol and are generally classified as wines, beers, and spirits. The latter contain the highest alcohol content and derive from the distillation of fermented products or maceration of raw material in ethyl alcohol. As ethanol can inhibit the activity of monoamine oxidase (MAO) and diamine oxidase (DAO), a high BA content in such beverages can represent a health hazard delaying the decomposition of these compounds in the human body. Another metabolite of wine named acetaldehyde as well as the anti-depressive drugs based on MAO inhibition, can cause the same interferences in BA metabolism [5].

### 2.1. Biogenic Amines’ Formation during the Winemaking Process

The formation of BAs in wine is associated with the amino acid content, the variety and degree of ripeness of grapes, and the vinification techniques. They can be naturally present in the raw material or formed during the different phases of production (Figure 1) and/or storage. Some stress factors, such as intensive nitrogenous fertilization, mold infections, damages by insects, and other growing conditions that are influenced by climate or soil type and composition, can increase the BA content of grapes. However, the BA levels detected in musts are usually low as they are produced and then degraded during winemaking. The primary fermentation by yeasts (especially *Saccharomyces cerevisiae*) and the malolactic fermentation of LAB are the main processes favoring BA formation [6].

The alcoholic fermentation (AF) is mainly carried out by yeasts converting sugars into ethanol, carbon dioxide (CO_2_), and other minor metabolites. The dominant *S*. *cerevisiae* strains, but also other species known as wild yeasts, i.e., *Saccharomyces bayanus*, *Brettanomyces bruxellensis*, and *Kloeckera apiculata*, have been shown to produce Bas, such as histamine, tyramine, spermidine, ethanolamine, agmatine, phenylethylamine, and cadaverine. The non-*Saccharomyces* yeasts rapidly decrease due to the strong selective pressure exerted by *S*. *cerevisiae* [7,8]. The AF can occur spontaneously or by the inoculation of specific strains and the most common are commercial starters from *S*. *cerevisiae* to ensure a reproducible, predictable, and controlled fermentation. Wines produced under this practice show low variability, complexity and typicity, and similar sensory properties. By contrast, the spontaneous fermentation may cause some problems to predict their evolution but wines have greater complexity and present higher differentiating characteristics. Moreover, the variety of yeast species within the winemaking environment is associated with the cellar resident population on wall surfaces, equipment, and oak barrels that play an important role in the wine aroma complexity. Thus, the cellar cleaning and hygienic practices can influence the winery microbiota affecting their diversity, composition, and evolution [9].

The malolactic fermentation (MLF) is the second important step in winemaking that is carried out by LAB species. In this process, the malic acid is converted in lactic acid and CO_2_, but also undesirable metabolites, such as BAs can be formed. The most often isolated LAB in wines, musts, and grapes belong to the genera *Oenococcus*, *Pediococcus*, *Lactobacillus*, and *Leuconostoc* [10]. They can be homofermentative, producing exclusively lactic acid and CO_2_ from sugars, or heterofermentative, producing also ethanol, acetic acid, and CO_2_. The MLF can occur by spontaneous fermentation due to indigenous LAB strains or by controlled fermentation after inoculation with selected starter cultures, mainly *Oenococcus oeni*. The latter is able growing with high alcohol and SO_2_ content, as well as with low pH and temperature. Other strains of the bacterial species of LAB producing BAs in wines are *Lactobacillus buchneri*, *Lactobacillus brevis*, *Lactobacillus hilgardii*, and *Leuconostoc mesenteroides*. Moreover, *Pediococcus damnosus* and *Pediococcus parvulus* have been reported as histamine producers [7]. Finally, the acetic acid bacteria (AAB) are considered spoilage microorganisms showing a strictly aerobic metabolism by which they oxidize ethanol into acetic acid by the acetaldehyde pathway [9].

Some other enological practices can also affect the final content of BAs in wine, such as SO_2_ concentration, clarification treatments, microorganism strains, and the aging period [6,11]. A pH value between 3.5 and 3.6 is considered critical to control microorganisms that can develop BAs in wine [12]. Some winemaking practices can also be used to hurdle BAs production, such as the good management of must pH and temperature, the antimicrobial stabilization with SO_2_, or the use of malolactic bacteria strains that have been screened for the absence of BA-producing genes [13].

The most BAs in red and white wines reported in literature are shown in Table 2. Histamine, tyramine, and putrescine showed the highest concentrations of up to 28, 37, and 122 mg/L in red wines, respectively. A great variability of BA amounts is generally found between white and red wines due to the maceration with the grape skins that occurs only in the red wine production that gives origin to high amounts of polyphenols and free amino acids. Moreover, the high fermentation temperature together with MLF, which in the white wine production has a short duration or is completely absent, are further conditions determining a higher BA formation in red wines [3].

### 2.2. Biogenic Amine in Beer and Other Alcoholic Beverages

Beer is a carbonated alcoholic beverage consumed worldwide. The most common sources of BAs in beer are the raw material (especially malt) and the contaminating microflora (mainly LAB). Histamine, cadaverine, putrescine, 2-phenylthylamine, and tyramine are the most detected BAs produced by decarboxylase positive microorganisms, while spermidine, 2-phenylethylamine, and spermine can be naturally found in malt, wort, and hops. Putrescine can show both origins. Moreover, the prolonged storage of the product can favor BA formation in beer [37]. With regards to the raw material, malt generally shows higher BA levels than hops, especially putrescine, spermine, spermidine, and agmatine. The barley variety and germination intensity can affect the amine concentrations. However, also in brewing, the fermentation phase is responsible for most BA formation. While yeast do not seem to influence their levels, many LAB (i.e., *Pediococcus damnosus*, *Lactobacillus frigidus*, *Lactobacillus brevissimilis*, and *L. brevis*) have been described as amine-producing microorganisms in beer. Finally, BAs can also be formed during storage [38]. Lorencová et al. [37] reported an increase of the detected BAs in relation to the progress of the storage period, with the histamine content above 20 mg/L at the end of the best-before date. Other abundant detected BAs were putrescine and cadaverine at levels <10 mg/L or ranging within the interval of 10–20 mg/L. However, the most frequently detected BAs in all tested samples with the highest concentrations was tyramine, which ranged from <10 to 20–50 mg/L. Then, 18% of the beer samples showed a total BA amount higher than 100 mg/L, demonstrating that they could be considered hazardous for consumers. In Table 3, some BAs found in beer samples are reported, while the production process is shown in Figure 2.

Other types of wine are the so-called fruit wines which are fermented alcoholic beverages made from a variety of ingredients other than grapes and/or additional flavors from fruits, flowers, and herbs. Płotka-Wasylka et al. [12] analyzed several fruit wines produced from different types of fruits such as apple, black lilac, quince, etc., and found histamine, dimethylamine, putrescine, cadaverine, methylamine, ethylamine, and butylamine at different concentrations based on the fruits used for their production. Cider is a slightly alcoholic beverage obtained by a reconstituted apple juice fermentation process or from apple must. Some traditionally manufactured ciders are produced by spontaneous fermentation carried out by indigenous microflora (mainly yeasts) but also through malolactic fermentation by LAB. Some strains belonging to the genera *Oenococcus*, *Lactobacillus*, *Pediococcus*, and *Enterococcus*, are often able to produce BAs including tyramine, putrescine, and cadaverine [39]. Rice wine, a traditional Chinese alcoholic drink, can contain high levels of BAs (e.g., putrescine, cadaverine, histamine, tyramine, and phenylethylamine) due to its great content of free amino acids deriving from the raw material during both the fermentation and aging process [40]. It is obtained by soaking raw rice for a long time and the soak water is then added to the must for fermentation. Some studies reported high BA levels in different Chinese rice wines, ranging from 5.0 to 78.5 mg/L for histamine or between 1.2 and 32.3 mg/L in semi-dry Chinese rice wine, and 6.2–37.6 mg/L in semi-sweet Chinese rice wine for tyramine [41].

Mead is an alcoholic beverage produced by the fermentation of honey and water. Silva et al. (2020) [42] detected putrescine, spermidine, and spermine in mead samples obtained from *Apis mellifera* honey as well as spermine and tyramine in those produced with honey from *Melipona quadrifasciata anthidioides* by using different finings and storage systems in oak barrel. In greater detail, when fruit peels (i.e., banana/*Apis* and passionfruit/*Melipona*) were used as finings, a BA increase was observed only in the first situation, i.e., the above-mentioned BAs were found in banana/*Apis* mead probably derived from the banana peel, while only spermidine was detected in the control samples. By contrast, the mead from passionfruit/*Melipona* did not show differences for BA content with the control group and therefore the use of this fining did not generate any new compounds. Liqueurs are spirit beverages containing alcohol and sugar at different percentages and obtained through different methods, i.e., through the infusion of herbs or roots in hot water and alcohol, cold maceration of fruits or flowers with distilled alcohol, or distillation of alcohol and flavoring agents. Various industrial and homemade liqueurs made from fruits, herbs, coffee, honey, and milk were analyzed by Cunha et al. (2017) [43] for BA presence. The coffee liqueurs showed the highest total content (2.1 mg/L), followed by honey liqueurs (1.9 mg/L), fruit liqueurs (1.3 mg/L), and herb liqueurs (1.0 mg/L). A significant difference was observed between BA concentrations in industrial and homemade liqueurs from fruits and herbs, and the latter showed a higher content. Ethylamine, morpholine, and cadaverine were found in all samples, while another six BAs (methylamine, dimethylamine, 1,3-diaminopropane, putrescine, histamine, and tyramine) were found in more than 50% of the investigated liqueurs.

**Table 3 foods-11-00353-t003:** Biogenic amines’ content (min–max values) in beers.

Biogenic Amines (mg/L)								Reference
Put	Cad	His	Spd	Spm	Tyr	Phe	Tryp	
2.1–12.8	0.2–1.4	nd-0.3			0.4–5.9	tr-0.2		Almeida et al., 2012 [44]
1.6	1.0	0.9			2.1			Matsheka et al., 2013 [45]
0.3–1.4	0.1–0.3	nd-0.6	0.2–0.8	0.2–0.7	nd-0.5	nd-0.5	0.3–2.6	Aflaki et al., 2014 [46]
2.1–12.8	0.2–1.4	nd-0.3			0.4–5.9			Ordóñez et al., 2016 [47]
nd-100.0	nd-100.0	nd-28.6	nd-50.0	nd-50.0	nd->100.0	nd-8.6	nd-28.6	Pradenas et al., 2016 [48]
1.6–4.1	0.4–0.8	nd			0.1–58.3	nd-0.4	nd	Redruello et al., 2017 [49]
3.6–8.9	0.0–1.3	nd-5.7	0.0–4.0		0.9–6.5	nd-0.3	0.1–0.5	Poveda, 2019 [50]
22.4–72.2	tr	nd	nd		tr-1.1	tr-1.1	nd	Angulo et al., 2020 [31]
nd-11.4	nd-12.7	nd	nd	nd	nd	nd-6.3	nd-4.1	Bae et al., 2020 [51]
1.5–8.2	0.2–1.4	tr-0.8	-	-	0.4–6.0			Bertuzzi et al., 2020 [52]
3.2–7	0.6–1.1	<LOQ-0.3	<LOQ-1.4			<LOQ->0.2		Díaz-Liñán et al., 2021 [53]
4.0–19	0.3–3.6	0.1–5.0	0.2–5.1	<LOQ-0.8	0.4–31.7	<LOQ-1.0	<LOQ-76.6	Nalazek-Rudnicka et al., 2021 [54]
0.3	0.3	0.2	0.3	0.6	0.7	0.5	0.6	Gil et al., 2022 [34]

Legend: nd = not detected; tr = traces; LOQ = limit of quantification; Put = Putrescine; Cad = Cadaverine; His = Histamine; Spd = Spermidine; Spm = Spermine; Tyr = Tyramine; Phe = β-Phenylethylamine; and Tryp = Tryptamine.

## 3. Biogenic Amine in Fermented Non-Alcoholic Beverages

Vinegar fermentation usually occurs after alcohol fermentation through the oxidation of ethanol by AAB. The traditional Bokbunja vinegar is produced from black raspberry through a two-step fermentation of alcohol and vinegar, and it is well known in Korea for its functional ingredients such as flavonoids, tannins, and phenolic compounds. During its production, many BAs can derive from both fermentation processes. Song et al. [55] isolated indigenous AAB strains with BA reduction ability, especially for histamine, from a total of 147 AAB-like strains from naturally fermented Bokbunja vinegar.

Coffee is one of the most popular drinks worldwide, ranking second only after crude oil. It can be produced by three different extraction techniques, i.e., decoction, infusion, and pressure. The most represented BAs in coffee beans and coffee beverages are putrescine, spermidine, spermine, and serotonin, while cadaverine and tyramine can also be present but at smaller concentrations [56]. The processing conditions of unripe coffee beans can affect the final levels of some BAs (i.e., spermine, spermidine, histamine, and cadaverine) as it seems that the de-pulping of unripe beans reduces fermentation and favors a uniform drying. By contrast, histamine, tryptamine, and cadaverine have been found in coffees of low cup quality and in defective coffee beans. Finally, the roasting effect on BA content is still controversial, as some authors reported their decrease after such process, while other studies showed that when it was stronger, the total BA levels were higher [57].

Another stimulating beverage that is very much consumed worldwide is tea. According to its production, it can be classified as unfermented (green tea), semi-fermented (oolong tea), fully fermented (black tea), or post-fermented (pu-er tea). To obtain green tea, the tea (*Camellia sinensis*) leaves undergo heating or steam treatment and fast drying, while during black tea production, they are subjected to weathering, various rolling, and crushing and/or tearing processes, followed by enzymatic maturation, and then final drying. By such a way, some enzymatic and oxidative reactions are responsible for BA formation, achieving levels of 20 and 14 mg/L for histamine and cadaverine, respectively [58].

## 4. Biogenic Amines in Non-Fermented Beverages

The non-fermented beverages are generally alcohol-free and include fruit juices, dairy beverages, plant milks, and soft or energy drinks. They can derive from foods of animal origin, especially milk, or fruits and vegetables (i.e., fruit juices, vegetable drinks made from soy, almond, rice, etc.). The presence of BAs in such drinks can be linked to the food matrix of origin (polyamines such as spermidine and spermine naturally derive from their precursor putrescine) or to inadequate hygienic conditions and/or microbial contamination [4]. The vegetable drinks are water-based beverages obtained from cereals, pseudo-cereals, oil seeds, legumes, or fruits. Their consumption is increasing in Europe and North America due to cow milk allergies or intolerances, but also in vegan diets. The production process involves an extraction with cold or hot water, a separation of solid fractions, and pasteurization or UHT treatment. Several BAs (histamine, serotonin, spermine, spermidine, putrescine, β-phenylethylamine, cadaverine, and tyramine) were investigated in plant milk samples from cereals and pseudo-cereals, and the total concentrations ranged from 1.9 to 9.3 mg/L. Histamine was the most found BA (82% of the total), varying from 1.9 to 8.4 mg/L, with a mean content of 6.2 mg/L. Cadaverine was the second most present BA, followed by tyramine and spermine, at concentrations of up to 1 and with ranges of 0.1–0.6 and 0.1 mg/L, respectively [59].

According to the production process, fruit juices can be made from fresh fruits without flavorings, colors, or other added ingredients, while fruit nectars are obtained from the addition of water, with or without added sugars, to fruit juices [60]. As polyamines are ubiquitous compounds in plants, their presence can also be found in the derived beverages [4]. Several studies [21,60,61] revealed putrescine (0.0–61.0 mg/L), spermine (0.2–3.6 mg/L), and spermidine (nd-5.4 mg/L) presence in different fruit juices. Phenylethylamine, tyramine, tryptamine, and methylamine were detected in pear nectar, orange juice, apple juice, mango juice, pineapple, litchi juice, and grapefruit juice, whereas histamine was found only in orange juice at low concentrations (nd-0.3 mg/L). Some soft drinks such as the orange carbonated-based drinks reported a similar BA profile (putrescine, histamine, spermidine, and spermine) of the orange juices from which they were produced. In Table 4, some BA concentrations in various fermented and non-fermented beverages are reported.

## 5. Harmful Effects of Biogenic Amines and Prevention Measures

Biogenic amines can be both essential and harmful to human health. According to their physiological effects, they are classified in vasoactive and psychoactive amines. The former, i.e., tyramine, tryptamine, and histamine, are involved in blood pressure control, whereas the psychoactive amines (histamine, putrescine, and cadaverine) act on the synaptic transmission. The consumption of foods and beverages with high BA concentrations can cause some adverse effects such as migraine, headache, bowel disorders, and allergic reactions. The latter are known as scombroid poisoning and cheese reaction due to histamine and tyramine, respectively. Putrescine and cadaverine cause hypotension and bradycardia, besides potentiating the toxicity of other amines. Moreover, high amounts of putrescine, spermidine, and spermine can accelerate the development of cancers, as they have been found in tissues with high growing rates [67]. Such polyamines can also react with nitrite to form carcinogenic nitrosamines [5]. The detailed adverse effects of most BAs in humans are reported in Table 5.

The European Food Safety Authority proposed potential Acute Reference Doses for the following BAs, i.e., 50 mg for histamine in healthy individuals but below the detectable limits for those with histamine intolerance; 600 mg for healthy individuals; 50 mg for those taking third-generation MAO inhibitors drugs; or 6 mg for those taking classical MAO-inhibitor drugs. According to the Food and Drug Administration, the tolerable level of histamine corresponds to 50 mg/Kg [75]. Nevertheless, both in the European Union and the United States, maximum limits have been set only for histamine in fish and fish products. With regard to wines, different countries have established upper limits of 2 mg/L (Germany), 3.5 mg/L (Netherlands), 5 mg/L (Finland), 6 mg/L (Belgium), 8 mg/L (France), and 10 mg/L (Australia and Switzerland) [13]. The adverse reactions of BAs can cause health problems when their concentrations in foods and beverages are too high or the human detoxification capacity is inhibited by gastrointestinal diseases, genetic predispositions, and the consumption of alcoholic beverages such as wine and beer containing ethanol and acetaldehyde derivates inhibiting MAO activity. Based on these situations, preventive measures able to hurdle their formation are essential to protect the public health. According to the code of good vinification practices to minimize BA formation in wine [76], some recommendations are focused on the selective grape harvest, aiming at eliminating bunches or parts of bunches that are damaged by fungi, avoiding any delay in the transport to the cellar. Particular attention must be paid on the maceration period, the pH increase, the addition of SO_2_ or lysozyme to control undesirable LAB in the musts, and the use of clarification products such as bentonite and fining agents. During clarification, some nutrients and suspended particles that can favor the growth of bacteria are removed. Then, microbiological analyses able to check yeasts and LAB population in must and wine should also be performed. Finally, good hygiene practices, temperature control, and proper storage are the main prevention actions for all beverages, other than wine, in which BAs can be formed. As histamine is generally not found in the raw material intended for beer production, it can be considered a good indicator of brewing hygiene and the bacterial contamination during all the production phases should be regularly controlled to ensure that the final product meets all quality and safety standards. Moreover, the under-pasteurization of beer can also increase the BA content [38]. In non-fermented beverages such as fruit juices, the hygiene of fruits and good manufacturing practices—pasteurization included—can prevent BA formation due to microbial growth before, during, and/or after production. However, further studies are needed to investigate the BA profile associated with the botanical origin [60].

## Figures and Tables

**Figure 1 foods-11-00353-f001:**
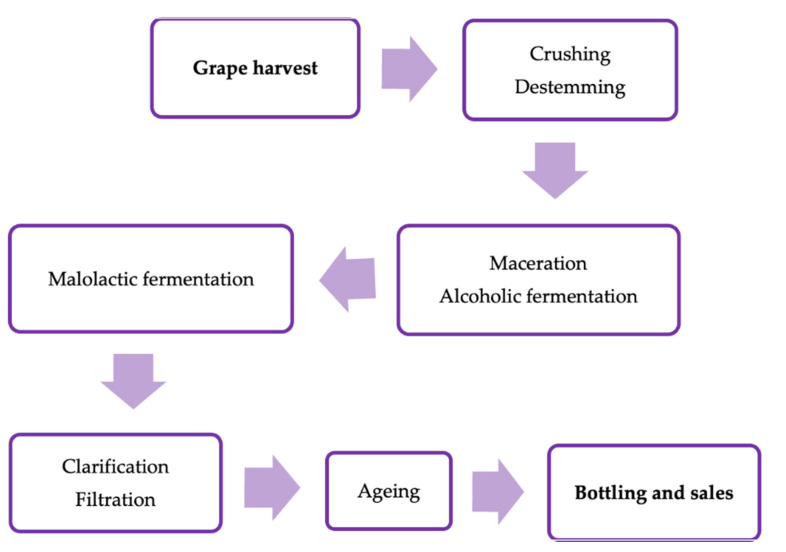
Flow sheet of vinification process.

**Figure 2 foods-11-00353-f002:**
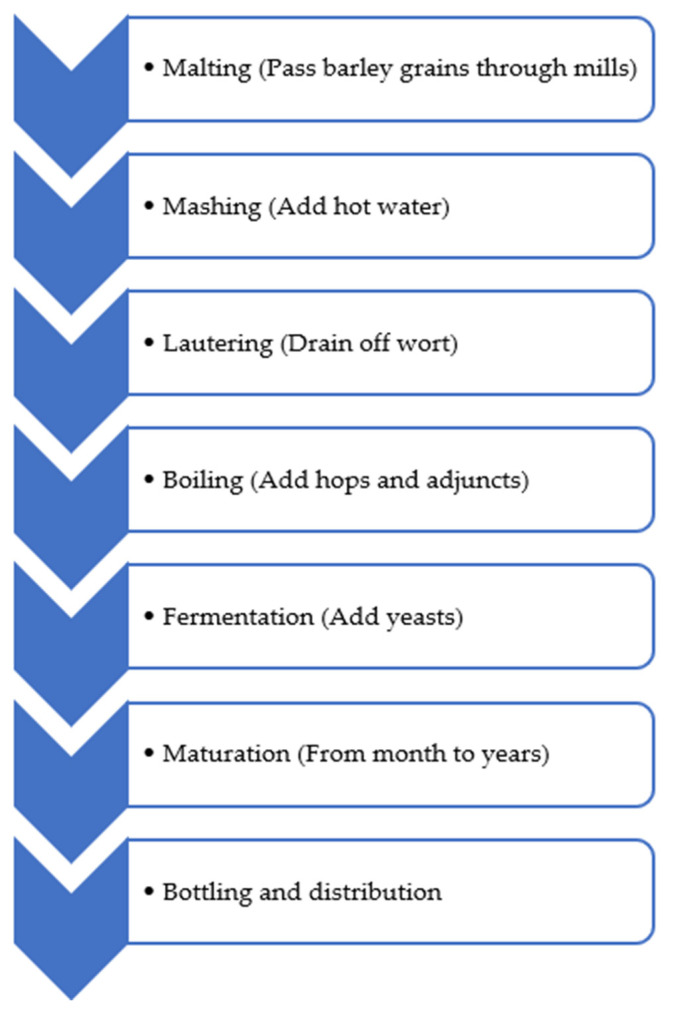
Main steps required for beer production.

**Table 1 foods-11-00353-t001:** Formula and chemical structure of most common BAs in beverages.

Biogenic Amine	Formula	Chemical Structure	Molecular Weight (g/mol)
Aliphatic	Aromatic	Heterocyclic
Histamine	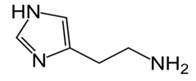			♦	111.15
Tyramine	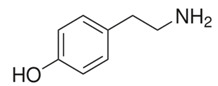		♦		137.18
Putrescine	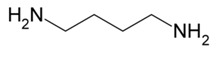	♦			88.15
Cadaverine	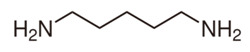	♦			102.18
Spermidine	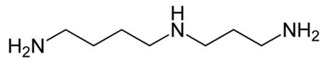	♦			145.25
Spermine	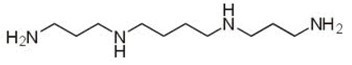	♦			202.34

**Table 2 foods-11-00353-t002:** Biogenic amines amount (min–max levels) in red and white wines.

Wine	Biogenic Amines (mg/L)										Reference
	Met	Eth	Put	Cad	His	Spd	Spm	Tyr	Phe	Tryp	
**Red**	nd	1.7–8.0	7.6–35.7	nd	nd-18.7			1.1–17.8		nd-16.2	Arrieta and Prats-Moya, 2012 [14]
	0.4–36.6	1.7–10.5	3.7–48.7	0.1–1.8	<0.5–14.1			<0.1–12.4	<0.1–2.7		Bach et al., 2012 [15]
			2.9–122.0		0.5–26.9			1.1–10.7			Konakowky et al., 2011 [16]
			7.1–19.0		2.2–16.2			0.5–37.3			Comuzzo et al., 2013 [17]
		0.8–6.5	2.4–31.8	0–1.1	0–10.8	nd	nd	0–18.8	nd		Martuscelli et al., 2013 [18]
	0.1	1.1	1.5	0.1	23.1	nq			0.2		Ramos et al., 2014 [19]
	0.2–1.7	4.1–11.3	10.2–32.8	0.6–2.4	tr-8.1	nd-1.3	nd	2.9–11.5	tr-1.2	tr-0.1	Tuberoso et al., 2015 [20]
	nd-1.4	0.7–1.9	3.8–11.1	0.5–1.6	nd-1.0	nq-0.7	nq-1.1	0.7–2.0	0.2–1.1	nd	Preti et al., 2015 [21]
					7.1–11.9			2.5–3.6	nd-2.6	nd-2.9	Jastrzębska et al., 2016 [22]
			8.2–16.2		4.3–12.3	3.1–15.9	3.4–21.0	2.1–14.5	nd-3.9		Restuccia et al., 2017 [23]
	<0.1	<0.2	0.8–7.5	<0–2.0	<0–2.3	<0–0.5	<0–0.3	<0–2.0		<0–1.2	Mitar et al., 2018 [24]
	<0.1–1.6	<0.2–1.2	<0–3.8	<0–0.5	<0–9.6	<0–6.1	<0–3.6	<0–3.0		<0–9.2	
			2.3–5.0	0.8–2.1	nd-1.0			nd-0.5			Diez Ozaeta et al., 2019 [25]
			24.8–34.2	1.1	nd	1.3–2.5	1.8	1.8–4.3			Esposito et al., 2019 [26]
			10.0	1.7	2.4			3.4			
		0.7–10.4	1.8–82.1	0–31.7	0–28.1	0–1.8	0–8.4				Filipe-Ribeiro et al., 2019 [27]
		0.9–10.4	1.8–82.1	<LOQ-20.5	<LOQ-28.1	<LOQ-1.6	<LOQ-6.9				Milheiro et al., 2019 [28]
			5.9–42.6	nd-4.3	nd-10.3			nd-4.1	nd-0.2	nd	Palomino-Vasco et al., 2019 [29]
			5.1–16.7	nd-2.2	1.1–9.1			nd-7.2	nd-0.5	nd	
			2.0–14.1	0.1–3.0	0.1–7.1			0.1–8.4			Žurga et al., 2019 [30]
			24.8–34.2	tr-1.1	tr	1.3–2.5		1.8–4.3	tr-3.5	nd	Angulo et al., 2020 [31]
			4.8–5.3	0.7	0.4–0.6	0.6–1.0		0.4–1.2	0.3		Rodríguez-Nogales et al., 2020 [32]
			nd-10.5	nd-3.4	nd-7.6	nd-1.3	nd-1.6	nd-6.6	nd-3.8	nd-2.5	Vinci et al., 2021 [33]
	0.3	0.4	0.4	0.4	0.8	0.7	0.8	2.3	0.5	1.1	Gil et al., 2022 [34]
**White**		1.1–8.6	0.8–12.8	0.3–1.2	0.3.4	nd	nd	0–6.8	nd		Martuscelli et al., 2013 [18]
	0.2–0.4	0.5	0.2–0.3	nq-0.1	2.8–8.9				nq-0.1		Ramos et al., 2014 [19]
	0.4–2.2	1.2–6.6	1.5–10.6	0.5–2.5	nd	tr	nd	tr	nd-1.8	tr-0.1	Tuberoso et al., 2015 [20]
					nd-2.9			nd-1.7	nd-1.8	nd-1.5	Jastrzębska et al., 2016 [22]
	<0.1–1.4	<0.2	1.0–2.1	tr	<0.03–0.6	<0.03	<0–0.4	<0–0.4		<0–1.2	Mitar et al., 2018 [24]
	<0.1	<0.2–0.6	0.3–1.5	tr	tr	tr	tr	tr		<0–0.7	
	nd-7.0	0.7–4.2	2.8–25.3	0.2–1.1	tr-16.6	nd-0.2	nd	nd-6.0	0.2–2.4	nd-0.4	Tuberoso et al., 2018 [35]
			1.9	1.5	0.8	nd	nd	0.4			Esposito et al., 2019 [26]
					0.2–3.0			0.1–1.2	nd-3.8	0.1	Perestrelo et al., 2020 [36]
			nd-4.2	nd-4.2	nd-4.4	nd-1.0	nd-1.6	nd-3.7	nd-3.2	nd-1.3	Vinci et al., 2021 [33]
	0.3	0.2	0.3	0.3	0.3	1.0	1.0	0.7	0.6	0.8	Gil et al., 2022 [34]

Legend: nd = not detected; nq = not quantifiable; tr = traces; LOQ = limit of quantification; Met = Methylamine; Eth = Ethylamine; Put = Putrescine; Cad = Cadaverine; His = Histamine; Spd = Spermidine; Spm = Spermine; Tyr = Tyramine; Phe = β-Phenylethylamine; and Tryp = Tryptamine.

**Table 4 foods-11-00353-t004:** Range (min–max concentrations) of biogenic amines in non-fermented beverages.

Product	Biogenic Amines (mg/L)										Reference
	Met	Eth	Put	Cad	His	Spd	Spm	Tyr	Phe	Tryp	
Rice wine			nd-58.9	nd-29.9	nd-24.9	nd	nd	1.2–37.6	nd	nd	Zhong et al., 2012 [62]
Brewed coffee			0.5–1.6	1.5–9.1	nd	nd	nd	nd-19.7	nd-5.0	nd-20.2	Özdestan, 2014 [63]
Ground coffee			3.9–15.2	13.6–75.1	nd	nd	nd	22.5–99.6	nd-22.8	3.0–37.9	
Espresso coffee			0.6–2.3	0.2–1.8	0.2–1.6	0.5–1.2	nd-2.0	0.3–1.9	0.2–1.2	ni	Restuccia et al., 2015 [57]
Rice wine			1.1–59.9	0–0.7	0.2–9.6	0–0.2	0.3–1.5	0.4–37.1	0.1–3.8	nd-0.5	Lee et al., 2015 [64]
Fruit nectars	nd	nd-2.5	1.1–3.3	2.0–17.2	nd	1.3–3.0	1.5–3.6	nd	nd		Preti et al., 2015 [21]
Rice wine			nd-32.3	nd-63.5	nd-72.1			nd-41.4			Ordóñez et al., 2016 [47]
Vinegar			nd-3.2	nd-0.1	nd-0.3			nd-0.2			
Cyder			nd-12.3	-	nd-6.9			nd-5.0			
Orange juice			0.1-2.2	-	tr			tr			
Apricot juice	nd	nd	1.4-7.1	4.0-17.9	nd	2.0–2.5	1.2–2.5	nd	nd	ni	Preti et al., 2016 [60]
Peach 50% juice	nd	nd	1.4–3.2	2.0–10.1	nd	1.3–2.0	1.2–2.7	nd	nd	ni	
Peach 70% juice	nd	nd	2.3–3.6	4.1–6.1	nd	2.0–4.4	1.4–1.9	nd	nd	ni	
Pear 50% juice	nd	nd	1.1–2.7	1.9–8.3	nd	1.2–1.8	1.2–3.5	nd	nd	ni	
Pear 70% juice	nd	1.1–1.2	1.4–4.4	3.8–6.2	nd	1.9–2.7	1.2–1.4	nd	nd	ni	
Apple concentrate juice	nd	nd-0.4	0.6–1.7	0.6–4.3	nd	0.2–0.7	0.2–1.0	nd	nd	ni	
Pineapple concentrate juice	nd	0.2–1.7	1.5–2.0	nd-3.1	nd	2.6–5.4	1.5–3.2	nd	nd	ni	
Grapefruit concentrate juice	nd-1.2	6.2–13.0	7.2–20.8	0.4–2.3	nd	1.0–2.2	0.3–0.5	nd	nd	ni	
Orange concentrate juice	nd-2.7	24.0–38.6	34.7–61.0	nd	nd	2.0–3.7	0.4–1.4	nd	nd	ni	
Black tea infusion			8.4–10.2	nd-14.0	nd-20.0	6.5–10.8	nd-0.3	nd	nd-2.0	nd	Spizzirri et al., 2016 [58]
Green tea			10.3–14.6	nd	nd	6.3–10.4	nd-11.5	nd	nd	nd	
Tea drinks			nd-6.9	nd	nd	4.3–6.7	nd	nd	nd	nd	
Fruit liqueurs	tr-0.2	tr-1.0	tr-2.5	tr-0.1	tr-0.2			tr	tr		Cunha et al., 2017 [43]
Herbs liqueurs	tr-1.1	tr	tr	tr-0.1	tr-0.2			tr	tr		
Coffee liqueurs	tr-1.1	tr-0.1	tr-0.4	tr	0.0–0.2			tr	tr		
Honey liqueurs	0.1	tr	0.1–0.7	tr-0.2	tr			tr-0.1	tr		
Fruit wine	0–0.1	0–0.3	0–9.9	0–0.9	0–1.5		tr	0–4.0	tr	tr	Płotka-Wasylka et al., 2018 [12]
Cherry wine					0.7–1.6	1.0–1.2	4.4–7.3	2.8–4.1	0.6–1.0	0.8–1.4	Sun et al., 2018 [65]
Raspberry wine			nd-3.3	nd-1.1	0.6–2.4	nd-2.6		1.1–11.3	0.2–4.1		Li et al., 2020 [66]
Chinese rice wines			nd-58.9	nd-98.7	nd-78.5	nd-27.1	nd-33.6	nd-100.8	nd		Fong et al., 2021 [41]
Cider with low alcoholic content			nd-45.9	nd-42.9	nd-9.3	0.7–7.6	1.1–9.9	nd-47.5	nd-7.7	nd-1.1	Lorencová et al., 2021 [39]
Cider with high alcoholic content			nd-45.0	nd-19.9	nd-18.1	1.2–4.8	1.7–9.1	nd-47.3	nd-9.6	nd	

Legend: nd = not detected; tr = traces; ni = not investigated; Met = Methylamine; Eth = Ethylamine; Put = Putrescine; Cad = Cadaverine; His = Histamine; Spd = Spermidine; Spm = Spermine; Tyr = Tyramine; Phe = β-Phenylethylamine; and Tryp = Tryptamine.

**Table 5 foods-11-00353-t005:** Main toxicological effects of biogenic amines.

Biogenic Amine	Toxicological Reactions	References
Cadaverine and Putrescine	Bradycardia, hypotension, increased cardiac output, paresis of the extremities. Gastric or intestinal cancer illness. N-nitrosamine formation reacting with nitrites. Potentiation of histamine’s toxic effect.	Ladero et al., 2010 [68] Benkerroum, 2016 [69]
Histamine	Flushing, hives, rashes, swelling. Abdominal cramps, bloating, diarrhea, nausea, oral numbness, and thirst. Dizziness, faintness, headache, loss of sight, and tremor. Hypotonia, low blood pressure, myocardial disfunction, rapid and weak pulse, and shock. Dyspnea, rhinitis, respiratory distress, and sneezing.	Comas-Basté et al., 2020 [70] Visciano et al., 2020 [71]
Phenylethylamine	Attention deficit disorder, depression, hypertension, headache, migraine, epilepsy, schizophrenia, and Parkinson’s disease.	Borah et al., 2013 [72] Benkerroum, 2016 [69]
Spermine and spermidine	N-nitrosamine formation reacting with nitrites. Potentiate the toxicity of the other biogenic amines.	Vinci et al., 2020 [4]
Tyramine	Nausea and vomiting. Headache, migraine, neurological disorders, salivation, and tearing. Hypertension, peripheral vasoconstriction, and respiratory disorders.	del Rio et al., 2017 [73]
Tryptamine	High blood pressure, headache, vomiting, and perspiration.	del Rio et al., 2020 [74]

## Data Availability

Not applicable.

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
