# Peer review of "Update on Biogenic Amines in Fermented and Non-Fermented Beverages"

_foods, 2022, doi:10.3390/foods11030353_

Round 1
Reviewer 1 Report
The manuscript is well organised, well written, easy to read and focuses on the (more) recent publications in this field, thus, the word "update" in the title. It is a very good introduction also for readers not experienced in BA. There are only few minor comments:
pg2 line 64: "the anti-depressive drugs based on MAO inhibiton"
pg3, lines 93 ff.: it is interesting to read that this is the same issue as with starters in fermented sausages.
pg3, line 111: "species": better "strains of the bacterial species..."?
pg? in section 2.2, line 199: "Melipona" in italics
Table 2: unclear what "nd-<LOD" means for min-max. "n" and "<LOD" should mean the same?
unclear what "<LOQ->LOQ" means for min-max. ">LOQ" means the must be a numerical value, which should be reported
The page numbering is not consistent, and page breaks shold be checked, e.g. headers for Tab. 1,3 not on the same page as bodies of the tables.
Author Response
The manuscript is well organised, well written, easy to read and focuses on the (more) recent publications in this field, thus, the word "update" in the title. It is a very good introduction also for readers not experienced in BA. There are only few minor comments:
pg2 line 64: "the anti-depressive drugs based on MAO inhibiton"
The authors modified as suggested.
pg3, lines 93 ff.: it is interesting to read that this is the same issue as with starters in fermented sausages.
The authors agree with your comment.
pg3, line 111: "species": better "strains of the bacterial species..."?
The authors modified as suggested.
pg? in section 2.2, line 199: "Melipona" in italics
The authors modified as suggested.
Table 2: unclear what "nd-<LOD" means for min-max. "n" and "<LOD" should mean the same?
The authors modified the Table 2 reporting only "nd"
unclear what "<LOQ->LOQ" means for min-max. ">LOQ" means the must be a numerical value, which should be reported
The authors added the numerical values when > LOQ.
The page numbering is not consistent, and page breaks should be checked, e.g. headers for Tab. 1,3 not on the same page as bodies of the tables.
The authors modified the layout of the Tables and consequently the page numbering.
Reviewer 2 Report
Manuscript ID: foods-1556870
Title: “Update on biogenic amines in fermented and non-fermented beverages”
This review summarizes the main biogenic amines in fermented and non-fermented beverages, their concentrations, and the prevention measures able to reduce their presence.
The basic idea of the manuscript is good, and it could be of practical interest.
However, some aspects could be improved
General comments
The prevention part is poorly developed and only focuses on wine
A figure of chemical structures of some biogenic amines and their synthesis could improve the work
In wine, the formation of biogenic amines during the different steps and processes is described, however, in the rest of the beverages this description is not as detailed. For example, in the case of beer, the formation of biogenic amines during the brewing has been described (Pavel Kalac and KÅ™ížek. 2003)
There is a recent review of biogenic amines in alcohol-free beverages that overlaps with this part in this article (Vinci and Maddaloni, 2020).
Author Response
This review summarizes the main biogenic amines in fermented and non-fermented beverages, their concentrations, and the prevention measures able to reduce their presence.
The basic idea of the manuscript is good, and it could be of practical interest.
However, some aspects could be improved
General comments
The prevention part is poorly developed and only focuses on wine
The authors added some information regarding beverages other than wine.
A figure of chemical structures of some biogenic amines and their synthesis could improve the work
The authors added Table 1 with the chemical structure of BAs.
In wine, the formation of biogenic amines during the different steps and processes is described, however, in the rest of the beverages this description is not as detailed. For example, in the case of beer, the formation of biogenic amines during the brewing has been described (Pavel Kalac and KÅ™ížek. 2003)
The authors added some information on the formation of BAs in beer.
There is a recent review of biogenic amines in alcohol-free beverages that overlaps with this part in this article (Vinci and Maddaloni, 2020)
The authors checked their text for the parts overlapping with Vinci and Maddaloni, 2020.
Round 2
Reviewer 2 Report
The majority of the comments have been emended in the revised manuscript.
The article could be published in the current state